# Visual Large Language Models Exhibit Human-Level Cognitive Flexibility

## Abstract

Cognitive flexibility has been extensively studied in human cognition but remains relatively unexplored in the context of Visual Large Language Models (VLLMs). This study assesses the cognitive flexibility of state-of-the-art VLLMs (GPT-4o, Gemini-1.5 Pro, and Claude-3.5 Sonnet) using the Wisconsin Card Sorting Test (WCST), a classic measure of set-shifting ability. Our results reveal that VLLMs achieve or surpass human-level set-shifting capabilities under chain-of-thought prompting with text-based inputs. However, their abilities are highly influenced by both input modality and prompting strategy. In addition, we find that through role-playing, VLLMs can simulate various functional deficits aligned with patients having impairments in cognitive flexibility, suggesting that VLLMs may possess a cognitive architecture, at least regarding the ability of set-shifting, similar to the brain. This study reveals the fact that VLLMs have already approached the human level on a key component underlying our higher cognition, and highlights the potential to use them to emulate complex brain processes.

## 1 Introduction

Cognitive flexibility, a key component of executive function, is fundamental to human adaptability and problem-solving (Dajani & Uddin, 2015; Uddin, 2021). This ability to shift between mental sets or strategies in response to changing environmental demands is crucial for everyday functioning and has been extensively studied in cognitive psychology research (Ionescu, 2012). The prefrontal cortex (PFC) is known to be central to this cognitive process(Spellman et al., 2021; Funahashi & Andreau, 2013), facilitating goal-directed behavior and controlled processing.

Recent advancements in artificial intelligence (AI), particularly in visual large language models (VLLMs) (Chen et al., 2024a), have sparked a growing need to assess these systems' cognitive abilities using paradigms analogous to human cognitive assessment (Momennejad et al., 2024; Chang et al., 2024; Qu et al., 2024). State-of-the-art VLLMs such as GPT-4o (OpenAI, 2024), Gemini-1.5 Pro (Reid et al., 2024), and Claude-3.5 Sonnet (Anthropic, 2024) have demonstrated remarkable capabilities in processing and interpreting both textual and visual information, excelling in tasks that demand complex reasoning and contextual understanding.

Despite these achievements, the extent to which VLLMs exhibit cognitive flexibility, especially in tasks requiring set-shifting and adaptation to changing rules, remains largely unexplored. While these models have demonstrated impressive performance across diverse tasks, their ability to flexibly adapt to changing environmental demands has not been systematically evaluated. This gap in our understanding is particularly significant given the increasing integration of VLLMs into complex, dynamic real-world environments where adaptability is crucial.

The Wisconsin Card Sorting Test (WCST), developed in the 1940s (Berg, 1948) and refined over decades, has emerged as the gold standard for assessing cognitive flexibility in both clinical and research settings (Miles et al., 2021; Greve, 2001). Originally developed to evaluate PFC function, it requires participants to discover sorting rules based on feedback and then flexibly shift to new rules when the criteria change (see Figure 1). The test's sensitivity to PFC function has been consistently demonstrated through lesion studies(Jodzio & Biechowska, 2010), neuroimaging research(Lie et al., 2006), and clinical observations, cementing its status as a crucial tool in understanding the cognitive flexibility. While other measures of cognitive flexibility exist, the WCST's established validity make it a core benchmark for evaluating this fundamental cognitive capacity.

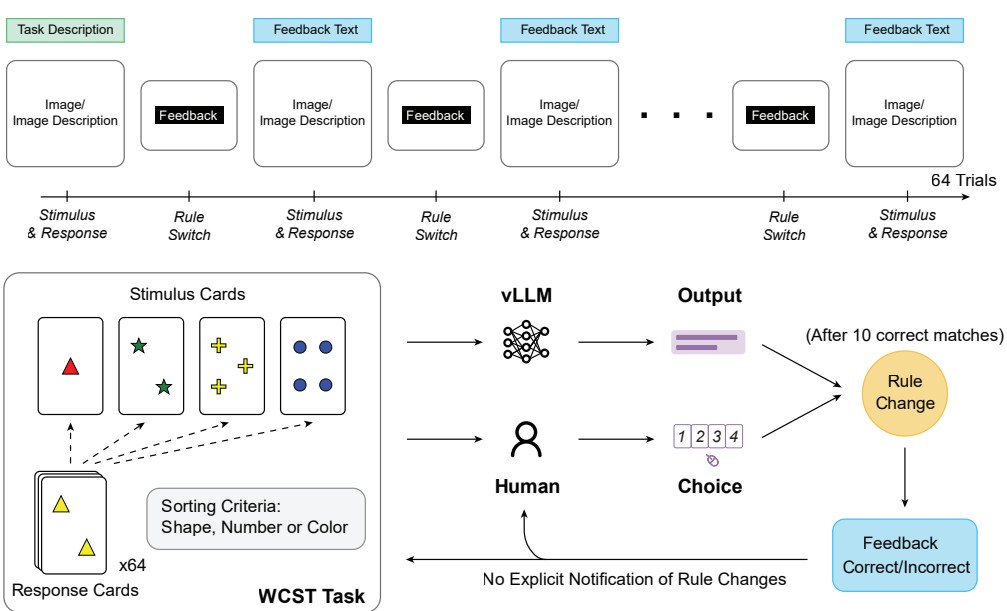

Figure 1: WCST Procedure and Sample Stimuli

This research aims to evaluate the cognitive flexibility in VLLMs using the WCST paradigm and investigate how different input modalities (image-based vs. text-based) ,prompting strategy (direct vs. chain-of-thought reasoning) and rule description specificity affect their performance. Additionally, we explore the potential of VLLMs to simulate specific patterns of cognitive impairment through role-playing, which enables us to understand human cognitive architecture. By comparing VLLM performance across varied conditions, we aim to elucidate their cognitive flexibility and inherent limitations. This investigation not only advances our understanding of VLLMs but also offers insights into the nature of cognitive flexibility itself.

## 2 RELATED WORK

### 2.1 COGNITIVE FLEXIBILITY AND ASSESSMENT METHODS

Neuroimaging studies have consistently implicated the PFC in cognitive flexibility tasks. The dorsolateral prefrontal cortex (DLPFC) and anterior cingulate cortex (ACC) play critical roles in set-shifting, with the DLPFC maintaining and updating task rules, and the ACC involved in conflict monitoring and error detection (Kim et al., 2011). The fronto-parietal network, encompassing these regions, dynamically reconfigures during flexibility-demanding tasks (Qiao et al., 2020). Cognitive flexibility is closely interrelated with other executive functions: working memory maintains task-relevant information and goals (Ionescu, 2012), while inhibitory control suppresses previous cognitive sets when rules change (Diamond, 2013).

Several tasks have been developed to assess cognitive flexibility in humans, with the WCST being a widely recognized measure of set-shifting ability (Eling et al., 2008). The WCST's sensitivity to PFC dysfunction has been extensively validated (Stuss & Levine, 2002). Complementary assessments include the Dimensional Change Card Sort (DCCS) task for children (Zelazo, 2006) and the computerized Intra-Extra Dimensional Set Shift (IED) subtest of the Cambridge Neuropsychological Test Automated Battery (CANTAB) (Heinzel et al., 2010), offering targeted measures across different populations and modalities.

## 2.2 Multifaceted Evaluation of LLMs

Recent studies have employed diverse assessments to evaluate large language models (LLMs) and VLLMs across various domains and tasks. Models like GPT-4 demonstrated human-level or superior performance on most theory of mind tests (Strachan et al., 2024). Similarly, research on human creativity found that ChatGPT-assisted ideas were more creative compared to those generated without LLM assistance (Lee & Chung, 2024). However, challenges persist in other areas. The Test of Time (ToT) benchmark exposed difficulties with complex temporal reasoning tasks, particularly those requiring multi-fact integration and intricate arithmetic operations (Fatemi et al., 2024). Despite strong performance on high-level vision tasks, state-of-the-art VLLMs struggled with basic geometric tasks that are straightforward for humans (Rahmanzadehgervi et al., 2024). A neuropsychological investigation revealed a discontinuous profile in ChatGPT's prefrontal functioning, with performance ranging from superior to impaired across different cognitive tasks (Loconte et al., 2023). To address the multifaceted nature of artificial intelligence, researchers have proposed new evaluation frameworks. A comprehensive framework for artificial general intelligence (AGI) tests inspired by cognitive science emphasizes the need for multidimensional intelligence assessment (Qu et al., 2024). Additionally, the concept of Turing Experiments (TEs) was introduced as a method for evaluating LLMs' ability to simulate human behavior in experimental settings (Aher et al., 2023).

## 2.3 Prompting Strategies

Prompting strategies significantly influence the performance of LLMs (Liu et al., 2023). The simplest approach, "Straight-to-Answer" (STA), directly queries the model without additional context. While effective for straightforward tasks, STA often falters on complex problems requiring multi-step reasoning. Chain-of-Thought (CoT) prompting encourages step-by-step reasoning(Wei et al., 2022), substantially improving performance on complex reasoning tasks (Chu et al., 2023). Variations such as zero-shot CoT (Kojima et al., 2022) and self-consistency CoT (Wang et al., 2022) have further refined this approach, adapting it to scenarios with limited or no task-specific examples. In multimodal contexts, visual CoT have extended these concepts to VLLMs (Chen et al., 2024b), demonstrating the potential for improved reasoning in tasks that combine textual and visual information. Other task-specific strategies, such as least-to-most prompting address challenges of easy-to-hard generalization (Zhou et al., 2022a), while meta-prompting and automatic prompt engineering techniques aim to optimize the prompts themselves (Pryzant et al., 2023; Zhou et al., 2022b).

## 3 Methods

### 3.1 Models and Experimental Procedure

This study focuses on three state-of-the-art VLLMs: GPT-4o, Gemini-1.5 Pro, and Claude-3.5 Sonnet. These models represent the current pinnacle of multimodal LLMs capabilities, demonstrating proficiency in processing both textual and visual inputs (see Appendix.A for details). We employs a standard version of the WCST-64 (Greve, 2001) to assess the cognitive flexibility of VLLMs. Our experimental design incorporates a 2x2 factorial structure, manipulating input modality (Visual Input (VI) / Textual Input (TI)) and prompting strategy (Straight to Answer (STA) / Chain of Thought (CoT)) to comprehensively evaluate VLLMs performance. This design resulted in four experimental conditions: STA-VI, STA-TI, CoT-VI, and CoT-TI. Each VLLM was tested independently across all four conditions, with 10 repetitions per condition. The arrangement of stimulus cards and the sequence of sorting rules were randomized for each repetition.

Additionally, we collected data from 30 cognitively healthy human participants (aged 20-35) as a baseline for comparison. Human participants interacted with a web-based interface designed to replicate the WCST experience while accommodating human response patterns (Figure B.3). The interface presented cards sequentially and allowed participants to indicate their sorting choices via button presses. The language used in instructions was carefully adapted to be more intuitive for human subjects while maintaining the essential structure of the task.

**Algorithm 1** WCST for VLLMs

1: **Initialize** $r \in \{\text{color, shape, number}\}$, $c \leftarrow 0$, $s \leftarrow 0$, $t \leftarrow 0$
2: Inform model: ”{Task description}”
3: **while** $t < 64$ **do**
4:   Present card and prompt for sorting (image/text for VI/TI, using STA/CoT)
5:   Record and parse model's response to extract selection
6:   **if** selection is correct **then**
7:     $c \leftarrow c + 1$
8:     **if** $c = 10$ **then**
9:       Change active rule $r$
10:       $s \leftarrow s + 1$
11:       $c \leftarrow 0$
12:     **end if**
13:   **else**
14:     $c \leftarrow 0$
15:   **end if**
16:   Provide feedback on correctness of current selection
17:   $t \leftarrow t + 1$
18: **end while**

We adapted the WCST for use with VLLMs while maintaining its core principles (see Algorithm 1 for the implementation). The test consists of a series of virtual cards, each featuring shapes (circle, cross, triangle, or star) in varying colors (red, green, yellow, or blue) and quantities (one to four). The models are tasked with sorting these cards according to an undisclosed rule (color, shape, or number), which changes periodically without explicit notification. The sorting rule changed after ten consecutive correct categorizations. The assessment concluded after 64 trials. Detailed descriptions of the task instructions are provided in Appendix B.1. Example stimuli for VI and TI conditions, and prompt templates for STA and CoT strategies are provided in Appendix B.2.

Data collection was fully automated using API calls to each VLLM. Model responses were recorded verbatim for each trial. Human participant data was collected through the web-based interface. All participants provided informed consent, and the study was approved by the institutional review board.

## 3.2 EVALUATION METRICS

Performance was primarily assessed using the following metrics which were chosen for their ability to quantify different aspects of cognitive flexibility:

Categories Completed (CC): The number of categories (sets of 10 consecutive correct sorts) completed.

$$CC = \sum_{i=1}^{n} I(c_i = 10) \tag{1}$$

where $n$ is the total number of trials, $c_i$ is the number of consecutive correct sorts at trial $i$, and $I(\cdot)$ is the indicator function.

Perseverative Errors (PE): The number of errors where the model persisted with a previously correct but currently incorrect rule.

$$PE = \sum_{i=1}^{n} I(r_i = r_{\text{prev}} \wedge r_i \neq r_{\text{current}}) \tag{2}$$

where $r_i$ is the rule used by the model at trial $i$, $r_{\text{prev}}$ is the previously correct rule, and $r_{\text{current}}$ is the current correct rule.

Non-Perseverative Errors (NPE): All errors that are not perseverative.

$$NPE = \text{Total Errors} - PE \tag{3}$$

NPE captures non-perseverative errors, potentially indicating exploration or random mistakes.

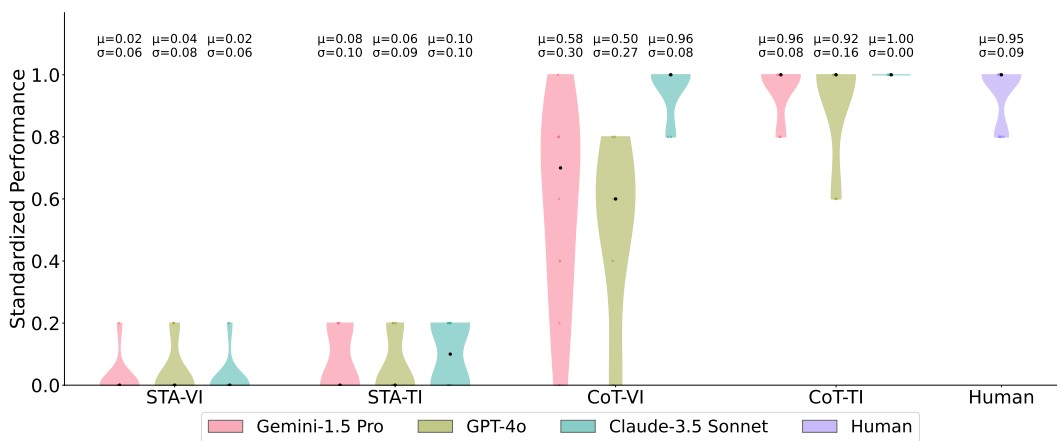

Figure 2: WCST Task Performance Across Models and Conditions

Trials to First Category (TFC): The number of trials required to complete the first category, indicating how quickly the model can deduce and consistently apply the first sorting rule.

$$TFC = \min\{i : c_i = 10\} \tag{4}$$

where $i$ is the trial number and $c_i$ is as defined in CC.

Conceptual Level Responses (CLR): The percentage of responses occurring in runs of three or more correct sorts, indicating conceptual understanding.

$$CLR = \frac{\sum_{i=1}^{n} I(c_i \geq 3)}{n} \times 100\% \tag{5}$$

where $I(c_i \geq 3)$ is an indicator function that equals 1 if the number of consecutive correct sorts up to and including trial $i$ is 3 or more, and 0 otherwise.

Failure to Maintain Set (FMS): The number of times the model makes an error after five or more consecutive correct sorts but before completing a category.

$$FMS = \sum_{i=1}^{n-1} I(5 \leq c_i < 10) \cdot I(c_{i+1} = 0) \tag{6}$$

where $i$ is the trial number and $c_i$ is as defined in CC.

These metrics collectively provide a comprehensive view of the VLLMs' cognitive flexibility(Schretlen, 2010), capturing various aspects such as rule learning, set-shifting, perseveration, and conceptual understanding.

## 4 RESULTS

### 4.1 WCST TASK PERFORMANCE ACROSS MODELS AND CONDITIONS

The WCST performance of GPT-4o, Gemini-1.5 Pro, and Claude-3.5 Sonnet exhibited marked variations across the four conditions (Figure 2). Their cognitive flexibility was measured using the CC metric, standardized on a 0-1 scale, with the human baseline($\mu = 0.95$, $\sigma = 0.09$). The CoT-TI condition consistently yielded superior outcomes across all VLLMs, followed by CoT-VI, STA-TI, and STA-VI, respectively, underscoring the critical influence of both prompting strategies and input modalities on VLLMs' set-shifting capabilities.

In the STA-VI condition, all VLLMs struggled significantly, with mean performances ranging from 0.02 to 0.04. The transition to STA-TI yielded modest improvements, particularly for Claude-3.5 Sonnet ($\mu = 0.10$, $\sigma = 0.10$). However, the introduction of CoT prompting precipitated a dramatic

Table 1: WCST Performance Metrics Across Models and Experimental Conditions

| Model | Condition | CC | PE | NPE | TFC | CLR (%) | FMS |
|---|---|---|---|---|---|---|---|
| Gemini-1.5 Pro | STA-VI | 0.10 (0.32) | 1.70 (5.38) | 40.00 (9.51) | 11.00 (-) | 5.00 (6.15) | 0.20 (0.42) |
| | STA-TI | 0.40 (0.52) | 10.10 (13.99) | 29.90 (17.90) | 19.25 (-) | 9.22 (7.88) | 0.10 (0.32) |
| | CoT-VI | 2.90 (1.60) | 7.40 (4.27) | 12.30 (12.37) | 19.11 (-) | 45.00 (18.95) | 0.50 (0.71) |
| | CoT-TI | 4.80 (0.42) | 6.80 (1.55) | 3.50 (1.72) | 13.30 (1.95) | 63.12 (4.67) | 0.10 (0.32) |
| GPT-4o | STA-VI | 0.20 (0.42) | 6.20 (13.21) | 32.70 (18.57) | 19.00 (-) | 10.31 (12.11) | 0.80 (1.48) |
| | STA-TI | 0.30 (0.48) | 11.60 (18.72) | 28.70 (19.81) | 12.00 (-) | 8.28 (5.90) | 0.30 (0.67) |
| | CoT-VI | 2.50 (1.43) | 7.60 (5.10) | 11.20 (10.03) | 17.38 (-) | 44.53 (15.51) | 1.10 (0.88) |
| | CoT-TI | 4.60 (0.84) | 7.60 (1.84) | 2.10 (0.88) | 12.60 (2.46) | 63.28 (5.86) | 0.10 (0.32) |
| Claude-3.5 Sonnet | STA-VI | 0.10 (0.32) | 3.10 (9.80) | 24.50 (10.97) | 17.00 (-) | 20.47 (12.60) | 1.60 (0.97) |
| | STA-TI | 0.50 (0.53) | 15.90 (17.53) | 22.90 (18.22) | 15.80 (-) | 8.90 (7.14) | 0.10 (0.32) |
| | CoT-VI | 4.80 (0.42) | 7.20 (2.82) | 2.20 (1.40) | 12.70 (1.57) | 65.16 (5.32) | 0.00 (0.00) |
| | CoT-TI | 5.00 (0.00) | 6.30 (0.82) | 2.00 (0.82) | 12.00 (0.94) | 67.50 (2.74) | 0.00 (0.00) |
| Human | STA-VI | 4.73 (0.45) | 6.87 (1.63) | 2.80 (1.69) | 12.93 (1.62) | 65.15 (4.35) | 0.10 (0.31) |

performance surge across all models. In the CoT-VI condition, Claude-3.5 Sonnet exhibited remarkable improvement ($\mu = 0.96$, $\sigma = 0.08$), while Gemini-1.5 Pro and GPT-4o also showed substantial gains. This stark contrast between STA and CoT conditions illuminates the pivotal role of explicit reasoning in augmenting VLLMs' cognitive flexibility.

The CoT-TI condition elicited peak performances, with Claude-3.5 Sonnet achieving perfection ($\mu = 1.00$, $\sigma = 0.00$), surpassing even the human baseline. Gemini-1.5 Pro ($\mu = 0.96$, $\sigma = 0.08$) and GPT-4o ($\mu = 0.92$, $\sigma = 0.16$) also demonstrated near-human or human-equivalent performance in this setting. Notably, the performance variability ($\sigma$) was generally higher in CoT conditions for Gemini-1.5 Pro and GPT-4o, indicating potential instability in their cognitive processes. The consistent superiority of TI over VI across all conditions suggests a potential advantage in processing textual over visual inputs.

The observed performance gradient, from near-chance levels in STA-VI to human-surpassing in CoT-TI, demonstrates the potential of VLLMs to exhibit human-like cognitive flexibility under appropriate conditions, while also highlighting the critical impact of prompting strategies and input modalities on their performance in tasks requiring set-shifting and rule adaptation.

## 4.2 DETAILED ANALYSIS BY EVALUATION METRIC

To offer a comprehensive assessment of the VLLMs' performance on the WCST, we analyzed six key metrics outlined in the previous section. Table 1 presents the mean scores and standard deviations across all evaluation metrics for each VLLM and condition. This analysis reveals distinct patterns in cognitive flexibility and set-shifting abilities among the models.

PE were most prevalent in the STA-TI condition for all models, with Claude-3.5 Sonnet showing the highest number of errors in this condition ($\mu = 15.90$, $\sigma = 17.53$). In the STA-VI condition, PE were relatively low for all models, as they largely failed to follow the rules at all. However, the transition to CoT conditions reduced PE, with Claude-3.5 Sonnet demonstrated the lowest number of PE in the CoT-TI condition ($\mu = 6.30$, $\sigma = 0.82$). This suggests that Claude-3.5 Sonnet may surpass human performance in adapting to changing rules, especially when provided with explicit reasoning prompts and textual descriptions.

NPE showed a dramatic reduction from STA to CoT conditions across all models, with improvements observed in the transition from VI to TI inputs. In STA conditions, NPE were extremely high, indicating near-random performance. The near-elimination of NPE in CoT-TI (e.g., Claude-3.5 Sonnet: $\mu = 2.00$, $\sigma = 0.82$) suggests that VLLMs can achieve a level of consistent rule application that exceeds human performance. This suggests that explicit reasoning prompts enable VLLMs to maintain a more consistent internal representation of the current sorting rule, reducing random errors.

All models required the fewest trials to complete the first category in the CoT-TI condition, with Claude-3.5 Sonnet performing best ($\mu = 12.00$, $\sigma = 0.94$), followed closely by GPT-4o ($\mu = 12.60$, $\sigma = 2.46$) and Gemini-1.5 Pro ($\mu = 13.30$, $\sigma = 1.95$). Notably, Claude-3.5 Sonnet outperformed the human baseline ($\mu = 12.93$, $\sigma = 1.62$).

CLR patterns showed substantial improvement from STA to CoT conditions for all VLLMs, with the highest percentages observed in the CoT-TI condition. Claude-3.5 Sonnet achieved the highest CLR in this condition ($\mu = 67.50\%$, $\sigma = 2.74\%$), followed by GPT-4o ($\mu = 63.28\%$, $\sigma = 5.86\%$) and Gemini-1.5 Pro ($\mu = 63.12\%$, $\sigma = 4.67\%$). This indicates that under CoT-TI conditions, VLLMs can maintain conceptual understanding at a level comparable to or exceeding human capability.

FMS were generally low in STA conditions, but this reflects the models' overall poor performance rather than true set maintenance. The transition to CoT conditions led to increased FMS in the VI condition, suggesting that improved overall performance paradoxically led to more instances of set loss after initial successful rule application. However, in the CoT-TI condition, Claude-3.5 Sonnet achieved perfect set maintenance (FMS = 0.00), outperforming the human baseline. This indicates that VLLMs can maintain exceptional consistency in rule application, potentially surpassing human capabilities in this aspect of cognitive flexibility.

These detailed metrics collectively reinforce the finding that CoT prompting, particularly when combined with textual inputs, substantially enhances VLLMs' cognitive flexibility as measured by WCST performance. While all models showed similar patterns across conditions, Claude-3.5 Sonnet consistently demonstrated superhuman cognitive flexibility in CoT-TI condition. The consistent pattern across all six metrics highlights the robustness of the effects of prompting strategy and input modality, while also revealing subtle differences in the cognitive capabilities of these VLLMs.

### 4.3 ANALYSIS OF INPUT MODALITY

To investigate the performance difference between visual and textual input conditions, we conducted a detailed analysis of each model's ability to accurately perceive and interpret the WCST card features. This analysis aimed to determine whether the performance gap was due to limitations in visual processing or differences in cognitive flexibility across modalities. We evaluated the models' accuracy in identifying the three key features of WCST cards: color, shape, and number, comparing their descriptions against actual card features for each trial (detailed in Appendix B.2).

The results indicate that all three models demonstrated high accuracy in visual feature recognition (Table 2). Claude-3.5 Sonnet demonstrated perfect accuracy across all features, while Gemini-1.5 Pro and GPT-4o showed a decline in visual capabilities, particularly when recognizing how many cards were present in the image and the number of shapes on each card. Notably, GPT-4o almost always misidentified 5 cards as 6 cards.

Table 2: Visual Feature Recognition Accuracy (%)

| Model | Count | Color | Shape | Number | Overall |
|---|---|---|---|---|---|
| Gemini-1.5 Pro | 75 | 100 | 100 | 97.81 | 96.97 |
| GPT-4o | 0 | 100 | 100 | 96.56 | 89.55 |
| Claude-3.5 Sonnet | 100 | 100 | 100 | 100 | 100 |

These findings suggest that the performance gap between VI and TI conditions is not solely attributable to limitations in visual feature extraction, but rather to the cascading effects of occasional visual misinterpretations on higher-order cognitive processes. In the VI condition, visual recognition errors can disrupt the model's ability to consistently apply a rule, necessitating re-exploration of the problem space. This phenomenon explains the increased variance observed in model performance under the CoT-VI condition compared to CoT-TI. The textual input's inherent precision eliminates this source of variability, allowing models to demonstrate more consistent cognitive flexibility. These results reveal the complex interplay between visual perception and executive function in VLLMs, highlighting the need for more robust visual processing pipelines.

### 4.4 IMPACT OF EXPLICIT RULE EXCLUSIVITY

All results in previous analyses were obtained under conditions that included both a general rule statement specifying " The correct answer depends on a rule, which will be based solely on either the number of shapes, the color of the shapes, or the shape type itself " and an explicit rule exclusivity constraint stating " There will be no combination of these characteristics to define the rule ". To

further investigate the robustness of VLLMs' cognitive flexibility, we conducted an additional study examining performance when explicit rule exclusivity was removed. We examined this under the CoT-TI condition, which had previously demonstrated near-human or superior cognitive flexibility for all models. The results are in Table 3.

Table 3: Impact of Explicit Rule Exclusivity on CC

| Model | Normal | w/o Constraints | Decline |
|---|---|---|---|
| Gemini-1.5 Pro | 4.8 (0.42) | 2.6 (2.01) | 2.2 |
| GPT-4o | 4.6 (0.84) | 3.5 (1.27) | 1.1 |
| Claude-3.5 Sonnet | 5.0 (0.00) | 4.7 (0.67) | 0.3 |

Gemini-1.5 Pro exhibited the most pronounced sensitivity to the absence of explicit rule exclusivity, with mean CC decreasing from 4.8 ($\sigma = 0.42$) with the constraint to 2.6 ($\sigma = 2.01$) without it, representing a 2.2 decline. GPT-4o demonstrated moderate sensitivity, with performance dropping from 4.6 ($\sigma = 0.84$) to 3.5 ($\sigma = 1.27$) categories. Claude-3.5 Sonnet showed the most robust performance, maintaining high functionality even without the explicit exclusivity statement, with only a marginal decline from perfect performance ($\mu = 5.0$, $\sigma = 0.00$) to near-perfect ($\mu = 4.7$, $\sigma = 0.67$). The observed increases in standard deviations across all models when the exclusivity constraint was removed indicate that explicit rule exclusivity not only enhances performance but also promotes more consistent cognitive flexibility.

The differential declines observed among models reflect disparities in their ability to maintain simple rule structures in the absence of explicit constraints against more complex possibilities. Claude-3.5 Sonnet's robust performance suggests a superior ability to infer and adhere to simpler rule structures, even when the possibility of more complex rules is not explicitly excluded.

## 4.5 SIMULATING COGNITIVE IMPAIRMENT

To explore the potential of VLLMs in modeling human cognitive impairment without modifying the models, we employed role-playing prompts to simulate three key aspects of the PFC function commonly impaired in various neurological conditions(Miller & Cohen, 2001; Stuss & Alexander, 2007): goal maintenance, inhibitory control, and adaptive updating. This method leverages the models' ability to imagine and simulate different cognitive states, allowing us to study how they conceptualize and perform under various impairment conditions without modifying the underlying model architecture.

We focused on the CoT-TI condition, as it consistently yielded the best performance across all models in our previous experiments. The specific role-playing prompts and analysis methods for this component are detailed in Appendix B.4. The results presented in Table 4, reveal that all three VLLMs demonstrated significant performance decrements under simulated impairment conditions, with patterns that align with neuropsychological observations of patients with prefrontal dysfunction.

Table 4: WCST Performance Under Normal and Simulated Impairment Conditions (CoT-TI)

| Model | Condition | CC | PE | NPE | TFC | CLR (%) | FMS |
|---|---|---|---|---|---|---|---|
| Gemini-1.5 Pro | Normal | 4.80 (0.42) | 6.80 (1.55) | 3.50 (1.72) | 13.30 (1.95) | 63.12 (4.67) | 0.10 (0.32) |
| | Goal Maint. (↓) | 1.90 (1.60) | 8.60 (7.83) | 12.00 (12.38) | 16.86 (-) | 37.03 (22.39) | 0.60 (0.70) |
| | Inhib. Ctrl. (↓) | 1.70 (1.49) | 6.40 (5.66) | 20.10 (14.04) | 32.00 (-) | 30.63 (18.91) | 0.90 (0.99) |
| | Adapt. Upd. (↓) | 3.90 (0.74) | 8.30 (2.71) | 6.70 (4.57) | 17.50 (6.20) | 56.09 (9.53) | 0.00 (0.00) |
| GPT-4o | Normal | 4.60 (0.84) | 7.60 (1.84) | 2.10 (0.88) | 12.60 (2.46) | 63.28 (5.86) | 0.10 (0.32) |
| | Goal Maint. (↓) | 3.50 (1.65) | 9.80 (2.66) | 4.50 (3.78) | 18.10 (9.46) | 52.34 (16.49) | 0.80 (1.03) |
| | Inhib. Ctrl. (↓) | 4.20 (1.03) | 10.10 (6.66) | 3.70 (2.41) | 14.00 (3.50) | 57.97 (11.66) | 0.30 (0.48) |
| | Adapt. Upd. (↓) | 4.30 (0.82) | 8.30 (3.13) | 2.80 (2.49) | 13.10 (1.79) | 61.56 (9.24) | 0.10 (0.32) |
| Claude-3.5 Sonnet | Normal | 5.00 (0.00) | 6.30 (0.82) | 2.00 (0.82) | 12.00 (0.94) | 67.50 (2.74) | 0.00 (0.00) |
| | Goal Maint. (↓) | 3.20 (1.40) | 12.50 (5.64) | 5.50 (4.79) | 17.50 (7.20) | 47.19 (17.44) | 0.60 (0.84) |
| | Inhib. Ctrl. (↓) | 1.50 (1.65) | 12.80 (13.82) | 18.70 (19.82) | 18.83 (-) | 23.59 (20.75) | 0.40 (0.52) |
| | Adapt. Upd. (↓) | 3.60 (1.26) | 8.60 (4.93) | 7.50 (9.35) | 20.60 (12.55) | 51.56 (14.91) | 0.00 (0.00) |

Gemini-1.5 Pro exhibited the highest sensitivity to simulated impairments, with substantial declines in CC across all conditions. The most severe impact was observed under inhibitory control impairment (CC reduced from 4.80 to 1.70), accompanied by a marked increase in NPE from 3.50 to 20.10.

GPT-4o demonstrated greater resilience, maintaining relatively stable performance across impairment conditions. The model's CC decreased modestly from 4.60 to 3.50-4.30, with PE showing consistent increases across conditions. Notably, NPE remained stable, indicating a robust ability to maintain overall response consistency even under simulated impairments. This stability suggests that GPT-4o's decision-making processes may be more resistant to perturbation.

Claude-3.5 Sonnet, despite showing the highest baseline performance (CC = 5.00), exhibited significant vulnerability to simulated impairments. The model showed increases in both PE and NPE under impairment conditions, particularly for inhibitory control (PE: 12.80, NPE: 18.70). This pattern suggests that Claude-3.5 Sonnet's high baseline performance may rely on finely tuned cognitive processes that are more susceptible to disruption when specific aspects of executive function are impaired.

Across all models, inhibitory control impairment consistently produced the most severe performance decrements and led to increased NPE and FMS, aligning with observations in patients with orbitofrontal damage(Stuss et al., 1983). Models frequently mentioned irrelevant card features, simulating distraction and impulsivity. Goal maintenance impairment primarily affected CLR and FMS, reflecting difficulties in consistently applying rules. This pattern is consistent with observations in patients with dorsolateral prefrontal cortex lesions (Stuss et al., 2000). Adaptive updating impairment had a more moderate impact, mainly affecting CC and CLR, while having less effect on FMS, consistent with difficulties in switching to new rules(Milner, 1963). These distinct patterns of impairment across models suggest that while VLLMs can simulate aspects of cognitive dysfunction, the underlying mechanisms of their decision-making processes may differ.

## 5 DISCUSSION

This study demonstrates that state-of-the-art VLLMs can achieve, and in some cases surpass, human-level cognitive flexibility as measured by the WCST, suggesting a potential for emulating and exceeding human set-shifting abilities in specific contexts. The observed performance gradient underscores the complex interplay between input modalities and prompting strategies. The performance gap between VI and TI conditions indicates that current VLLMs may rely more heavily on language-based reasoning pathways, even when processing visual information. Explicit reasoning prompts enable VLLMs to maintain more stable internal representations of task rules.

Our analysis of explicit rule exclusivity reveals a critical dependence of VLLMs on precise task instructions. The significant performance decline observed when specific rule constraints were removed highlights the models' reliance on explicit information to guide their decision-making processes. This finding suggests that VLLMs' impressive performance in structured tasks may not fully generalize to more ambiguous real-world scenarios without careful consideration of instruction design. The simulation of cognitive impairments through role-playing prompts demonstrates the potential of VLLMs to model complex patterns of executive dysfunction. The distinct performance profiles observed under simulated goal maintenance, inhibitory control, and adaptive updating impairments closely mirror patterns seen in human neuropsychological research.

Future research should focus on elucidating the underlying mechanisms that enable VLLMs to perform set-shifting tasks and investigating the generalizability of these abilities to other domains of executive function. Additionally, the development of more sophisticated visual processing capabilities and the integration of multimodal information processing warrant further exploration. The potential of VLLMs to simulate specific patterns of cognitive impairment also opens up new possibilities for creating realistic models of neuropsychological conditions, which could have applications in both clinical research and AI safety. By analyzing VLLMs' internal representations during simulated impairments, we could potentially decode the computational principles underlying various cognitive functions, complementing traditional neuroscience methods.

In conclusion, this study provides insights into our understanding of cognitive flexibility in VLLMs, revealing capabilities that match or exceed human performance and important limitations that de-

pend on task framing and input modalities. As these models continue to evolve, a deeper understanding of their cognitive processes will be crucial for harnessing their potential while addressing their constraints, ultimately leading to more adaptive and robust AI systems that can flexibly navigate complex, real-world environments.

# 6    LIMITATIONS

This study's findings are constrained by several factors. Our focus on a limited subset of current VLLMs may not generalize to the entire spectrum of models or future iterations. The role-playing approach to simulating cognitive impairments, while innovative, requires validation against clinical populations to confirm its fidelity. The observed sensitivity to input modalities and instruction specificity suggests potential variability in less structured scenarios. Our analysis of visual processing was limited to feature recognition, neglecting deeper aspects of visual cognition. Additionally, the study did not investigate the underlying mechanisms enabling VLLMs to perform set-shifting tasks, constraining our understanding of how these abilities emerge from the models' architecture and training. Addressing these limitations in future research will be crucial for a more comprehensive understanding of cognitive flexibility in VLLMs and their potential applications in complex cognitive tasks.

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

# A    MODEL DETAILS

This study employs three state-of-the-art visual language models: Gemini-1.5 Pro, GPT-4o, and Claude-3.5 Sonnet. Table A1 presents a comparative overview of these models' key characteristics.

Table A.1: Comparison of Visual Language Models

| Characteristic | | Gemini-1.5 Pro | GPT-4o | Claude-3.5 Sonnet |
|---|---|---|---|---|
| Multimodal Capabilities | | Text, audio, image, video | Text, audio, image | Text, image |
| API Version | | gemini-1.5-pro | gpt-4o-2024-05-13 | claude-3-5-sonnet -20240620 |
| Access Method | | aistudio.google.com | platform.openai.com | console.anthropic.com |
| Context Window | | 2M tokens | 128K tokens | 200K tokens |
| Maximum Out-Tokens | | 8,192 tokens | 4,096 tokens | 4,096 tokens |
| Knowledge Cutoff | | November 2023 | October 2023 | April 2024 |
| Release Date | | May 2024 | May 2024 | June 2024 |
| Model Ranking (till 2024-08-31) | LMSYS | #4 | #1 | #2 |
| | OpenCompass | #5 | #1 | #2 |
| | Benchmarks | #4 | #2 | #1 |

## A.1    GEMINI-1.5 PRO

Gemini-1.5 Pro, created by Google, employs a Mixture of Experts architecture, allowing for efficient processing of both textual and visual inputs. While specific architectural details are proprietary, the model demonstrates strong performance across various tasks. For image processing, Gemini-1.5 Pro uses a standardized approach where each image is equivalent to 258 tokens, regardless of size. Large images are scaled down to a maximum of 3072x3072 pixels, while small images are scaled up to 768x768 pixels, both preserving aspect ratio.

## A.2    GPT-4O

GPT-4o, developed by OpenAI, represents an advanced iteration of the GPT series. It utilizes a transformer-based architecture and incorporates visual processing capabilities. GPT-4o offers adaptive image processing with low and high resolution modes, allowing for a balance between processing speed and detail level. In low resolution mode, it processes a 512px x 512px version of the image, representing it with 85 tokens. The high resolution mode initially processes a low-res image, then creates detailed 512px x 512px crops, each represented by 170 tokens.

## A.3    CLAUDE-3.5 SONNET

Developed by Anthropic, Claude-3.5 Sonnet builds upon previous Claude models, incorporating enhanced visual understanding capabilities. The model utilizes a transformer-based architecture optimized for multimodal inputs. Claude-3.5 Sonnet balances multi-image processing by resizing images that exceed 1568 pixels on the long edge or approximately 1,600 tokens. It calculates token usage based on image dimensions (tokens = (width px * height px)/750) and emphasizes image clarity and text legibility for optimal performance.

# B    PROMPTS AND TOKEN USAGE

## B.1    DETAILED PROMPTS

The WCST setup consists of four stimulus cards, each featuring unique combinations of color (red, green, yellow, blue), shape (triangle, star, cross, circle), and number (one, two, three, four) of symbols. A series of 64 response cards is used, each sharing properties with the stimulus cards but in different combinations. The sorting rules are based on three possible categories: color, shape, or

number. In our implementation, each trial is presented to the VLLMs as a single image containing two rows. The top row displays the four stimulus cards, while the bottom row shows one response card. This format is consistent across all 64 trials, providing a standardized visual input for the models to process. For the text-based conditions (TI), detailed descriptions of these images are provided instead.

The test procedure begins with the VLLM being instructed to match each response card to one of the stimulus cards based on a rule that it must deduce from feedback. After each match attempt, the VLLM receives feedback (correct or incorrect) without explicit mention of the current sorting rule. The sorting rule changes after 10 consecutive correct matches, without notification to the VLLM. The test concludes after all 64 cards have been presented.

We implemented four distinct experimental conditions to assess the VLLMs' performance: STA-VI (Straight to Answer with Original Image input), STA-TI (Straight to Answer with Original Text description input), CoT-VI (Chain of Thought reasoning with Original Image input), and CoT-TI (Chain of Thought reasoning with Original Text description input). Figure B.1 provides a visual representation of the WCST procedure and sample stimuli used in our study.

The prompting strategies for each condition are illustrated in Figure B.2. For the CoT conditions, VLLMs were explicitly instructed to verbalize their reasoning process, including their observations, hypotheses about the current rule, and justification for their sorting decisions.

Figure B.3 presents the web-based interface developed for human participants. This interface was designed to closely mimic the experience of VLLMs while accommodating human interaction patterns. It features a clear presentation of stimulus and response cards, along with intuitive controls for participants to indicate their sorting choices. The interface also provides immediate feedback on sorting decisions, mirroring the feedback mechanism used with VLLMs.

To explore the VLLMs' capacity to simulate cognitive impairments, we introduced role-playing scenarios as described in Figure B.4. This figure outlines the specific instructions given to models for simulating various prefrontal cortex dysfunctions, including impaired goal maintenance, inhibitory control deficits, and adaptive updating impairments. These scenarios were carefully designed to mimic specific cognitive deficits commonly observed in neurological conditions, allowing us to assess the models' ability to flexibly adapt their behavior to simulate human-like cognitive impairments.

Figure B.5 provides a detailed example of a typical VLLM interaction during the WCST. This example illustrates how models process the presented cards, articulate their reasoning (in CoT-TI conditions), and make decisions. Figure B.6 showcases the visual processing capabilities of VLLMs.

## B.2 Visual Accuracy Calculation

The visual accuracy of VLLMs was assessed using a comprehensive scoring system that evaluated their ability to correctly identify key features of the WCST cards across 64 trials. The system encompassed five distinct measures: Card Count Accuracy, Color Accuracy, Shape Accuracy, Number Accuracy, and Overall Accuracy. For each trial, models were evaluated on their ability to correctly identify the presence of five cards and accurately describe the color, shape, and number of symbols on each card. Detailed descriptions of the Visual instructions are provided in Figure B.2. Figure B.6 showcases the visual processing capabilities of VLLMs.

Count Accuracy was calculated as the proportion of trials where the model correctly identified the presence of five cards:

$$ACC_{count} = \frac{\sum_{i=1}^{64} I(c_i = 5)}{64} \times 100\% \tag{1}$$

where $I(c_i = 5)$ is an indicator function that equals 1 if the model correctly counted 5 cards in trial $i$, and 0 otherwise.

Color Accuracy , Shape Accuracy , and Number Accuracy were calculated similarly, assessing the model's performance across all cards in all trials:

$$ACC_{feature} = \frac{\sum_{i=1}^{64} \sum_{j=1}^{5} I(f_{ij} = f_{ij}^*)}{64 * 5} \times 100\% \tag{2}$$

where $feature \in color, shape, number$, $f_{ij}$ is the model's identification of the feature for card $j$ in trial $i$, and $f_{ij}^*$ is the correct feature.

The Overall Accuracy ($ACC_{overall}$) was computed as a composite score, incorporating all correct identifications while applying a penalty for overcounting cards. First, we define a penalty function $P$ for overcounting:

$$P = \frac{\sum_{i=1}^{64} 0.5 \times \max(0, c_i - 5)}{64} \times 100\% \tag{3}$$

where $c_i$ is the number of cards counted by the model in trial $i$. This penalty deducts 0.5 points for each card counted beyond the correct number of 5 in any given trial.

The Overall Accuracy is then calculated as:

$$ACC_{overall} = ACC_{count} + \sum ACC_{feature} - P \tag{4}$$

This assessment of the VLLMs' visual processing capabilities enables detailed comparisons across models and features. By evaluating multiple aspects of visual perception, from basic counting to complex feature recognition, the system offered insights into the strengths and limitations of each model's visual cognition in the context of the WCST.

## B.3 TOKEN USAGE

To provide insight into the computational resources required, we list token usage across models and conditions (Table B.1). Across all models, VI conditions consistently required more tokens than TI conditions, reflecting the additional computational demand of processing visual information. CoT conditions consumed significantly more tokens than STA conditions, indicating the increased computational cost of explicit reasoning processes. Among the models, Claude-3.5 Sonnet showed the highest token usage across all conditions, suggesting a more computationally intensive approach to task processing. These token usage patterns provide valuable insights into the relative efficiency and resource requirements of different VLLMs and experimental conditions in cognitive flexibility tasks.

Table B.1: Token Usage and Cost Analysis

| Model | Condition | Task | Last Token Avg | Session Token Avg | Session Price Avg | Total Tokens | Total Price |
|---|---|---|---|---|---|---|---|
| Gemini-1.5 Pro | STA-VI | WCST | 18,221 | 604,260 | $2.12 | 6,042,605 | $21.17 |
| | STA-TI | WCST | 6,631 | 227,378 | $0.8 | 2,273,782 | $7.95 |
| | CoT-VI | WCST | 19,898 | 658,489 | $2.32 | 6,584,887 | $23.18 |
| | | WCST | 9,885 | 338,747 | $1.21 | 3,387,469 | $12.09 |
| | | WCST w/o restriction | 12,795 | 421,199 | $1.52 | 4,211,991 | $15.19 |
| | CoT-TI | WCST Goal Maint | 8,970 | 311,489 | $1.11 | 3,114,890 | $11.08 |
| | | WCST Inhib Ctrl | 10,447 | 355,556 | $1.27 | 3,555,564 | $12.74 |
| | | WCST Adapt Upd | 8,423 | 295,753 | $1.05 | 2,957,531 | $10.49 |
| GPT-4o | STA-VI | WCST | 7,373 | 251,712 | $1.26 | 2,517,120 | $12.62 |
| | STA-TI | WCST | 6,831 | 233,900 | $1.17 | 2,338,995 | $11.66 |
| | CoT-VI | WCST | 20,210 | 670,972 | $3.49 | 6,709,718 | $34.87 |
| | | WCST | 20,216 | 672,265 | $3.5 | 6,722,651 | $34.96 |
| | | WCST w/o restriction | 23,093 | 749,301 | $3.91 | 7,493,007 | $39.12 |
| | CoT-TI | WCST Goal Maint | 18,414 | 619,910 | $3.22 | 6,199,099 | $32.19 |
| | | WCST Inhib Ctrl | 18,642 | 624,323 | $3.24 | 6,243,230 | $32.43 |
| | | WCST Adapt Upd | 18,824 | 634,350 | $3.29 | 6,343,505 | $32.93 |
| Claude-3.5 Sonnet | STA-VI | WCST | 27,404 | 903,104 | $2.72 | 9,031,040 | $27.2 |
| | STA-TI | WCST | 7,073 | 242,113 | $0.73 | 2,421,131 | $7.34 |
| | CoT-VI | WCST | 43,704 | 1,426,502 | $4.48 | 14,265,023 | $44.78 |
| | | WCST | 19,257 | 641,037 | $2.08 | 6,410,367 | $20.77 |
| | | WCST w/o restriction | 20,718 | 675,461 | $2.2 | 6,754,606 | $21.96 |
| | CoT-TI | WCST Goal Maint | 23,806 | 778,877 | $2.54 | 7,788,771 | $25.42 |
| | | WCST Inhib Ctrl | 24,550 | 802,087 | $2.62 | 8,020,867 | $26.18 |
| | | WCST Adapt Upd | 24,774 | 799,338 | $2.62 | 7,993,378 | $26.16 |

**Task Description (Normal) :**

In this task, you will be presented with an image containing five cards. Each card has figures that differ in number, color, and shape. Four cards will be displayed at the top of the image, and one card will appear in the bottom left corner of the image. Your task will be to try to determine which of the four cards at the top the bottom left card belongs to. ***The correct answer depends upon a rule, which will be based solely on either the number of shapes, the color of the shapes, or the shape type itself.*** *There will be no combination of these characteristics to define the rule.* For example: If the rule is to match according to color, you should choose the card that matches the color of the bottom left card. If the rule is to match according to shape, you should choose the card that has the same shape as the bottom left card. If the rule is to match according to the number of shapes, you should choose the card that has the same number of shapes as the bottom left card. We will not tell you what the rule is. However, we will provide feedback after each selection, indicating only whether your selection was correct or incorrect, without any feedback into your reasoning process. The rule may change during the task. When the rule changes, you should figure out what the rule is as quickly as possible. When you make your selection, for each trial, respond ONLY in this format: Selection: [Card number, 1-4]. Example response: Selection: 2. You can just choose from the top four cards, not from the bottom left card. Feedback will be given after each selection, and you will be presented a different image. This task consists of 64 trials involving different images. Please proceed with the first image.

**Task Description ( w/o Constraints) :**

In this task, you will be presented with an image containing five cards. Each card has figures that differ in number, color, and shape. Four cards will be displayed at the top of the image, and one card will appear in the bottom left corner of the image. Your task will be to try to determine which of the four cards at the top the bottom left card belongs to. ***The correct answer depends upon a rule, which will be based solely on either the number of shapes, the color of the shapes, or the shape type itself.*** For example: If the rule is to match according to color, you should choose the card that matches the color of the bottom left card. If the rule is to match according to shape, you should choose the card that has the same shape as the bottom left card. If the rule is to match according to the number of shapes, you should choose the card that has the same number of shapes as the bottom left card. We will not tell you what the rule is. However, we will provide feedback after each selection, indicating only whether your selection was correct or incorrect, without any feedback into your reasoning process. The rule may change during the task. When the rule changes, you should figure out what the rule is as quickly as possible. When you make your selection, for each trial, respond ONLY in this format: Selection: [Card number, 1-4]. Example response: Selection: 2. You can just choose from the top four cards, not from the bottom left card. Feedback will be given after each selection, and you will be presented a different image. This task consists of 64 trials involving different images. Please proceed with the first image.

Figure B.1: WCST Task Description

**Image :**

**Task Description :**

In this task, you will be presented with an image containing five cards. Each card has figures that differ in number, color, and shape. Four cards will be displayed at the top of the image, and one card will appear in the bottom left corner of the image. Your task will be to ...

**Image Description :**

The image shows four cards with white background at the top, the first card is with 1 single red triangle, the second card is with 2 green stars, the third card is with 3 yellow cross signs, the fourth card is with 4 blue circles. And there is one card at the bottom left with 1 single blue circle.

## PROMPT_TYPES

**Straight-to-Answer (STA) :**

Provide your final answer immediately without explaining your reasoning. Your response should be concise and direct, containing only the essential information required by the task. Do not include any additional explanations or thought processes.

**Chain-of-Thought (CoT) :**

Let's think step by step, first explain your thought process in detail, then clearly state your final response to the task.

## VISION_QUESTION

First, state how many cards you see in the image. Then, describe each card in detail. The cards contain shapes that vary in color, type, and number. For each card, provide the following information: number of shapes, color of shapes, and shape type. Format your response as: Number of cards: [number], Card Label: [number, color, shape], ...

Figure B.2: WCST Prompt Description

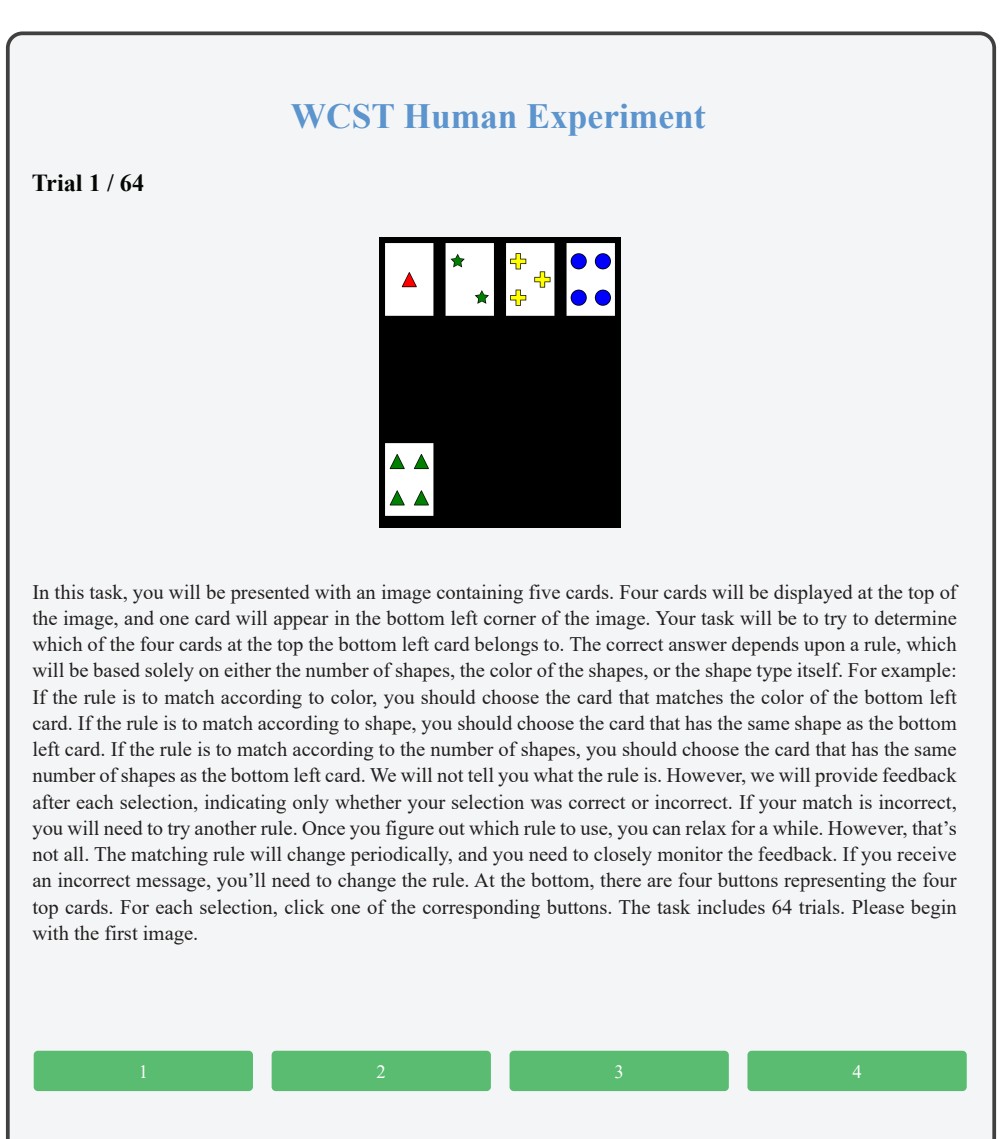

Figure B.3: WCST Human Experiment Interface

**ROLE_DESCRIPTIONS**

**Goal_Maint**

While performing this task, you have significant difficulty maintaining goals in your working memory. You often lose track of the current sorting rule you're supposed to be following. You struggle to consistently apply a rule across multiple trials. You may need to refer back to the instructions more frequently than usual. Despite these challenges, you still attempt to complete the task to the best of your ability. Your responses should reflect a genuine effort to perform the task, but with clear signs of difficulty in maintaining focus on the current goal or rule.

**Inhib_Ctrl**

As you work on this task, you have significant difficulty inhibiting irrelevant information and controlling impulsive responses. You're are easily distracted by unimportant details and may occasionally struggle to focus solely on the relevant aspects of the cards. You often make impulsive choices without fully considering all options. Despite these challenges, you still attempt to complete the task to the best of your ability. Your responses should reflect a genuine effort to perform the task, but with clear signs of difficulty in inhibiting irrelevant information and controlling impulsive reactions.

**Adapt_Upd**

During this task, you have significant difficulty in adapting to changes. You strongly tend to stick with a previously correct sorting rule, even when it's no longer effective. When faced with negative feedback, you struggle to recognize that a rule change has occurred and instead continue applying the old rule. Despite these challenges, you still attempt to complete the task to the best of your ability and you can eventually adapt after more attempts. Your responses should reflect a genuine effort to perform the task, but with clear signs of difficulty in adapting to changes and updating your behavior based on feedback.

Figure B.4: WCST Role Description

---

**TASK_RESPONSE_EXAMPLE**

**Task Description :**

In this task, you will be presented with an image containing five cards. Each card has figures that differ in number, color, and shape. Four cards will be displayed at the top of the image, and one card will appear in the bottom left corner of the image. Your task will be to ...

◆ **Gemini-1.5 pro (CoT-TI) :**

The bottom left card has one blue circle. Now I need to decide if the rule is based on number of shapes, color, or shape itself.
**Number:** If the rule is the number of shapes, the correct choice would be the first card (1 red triangle) because it also has one shape.
**Color:** If the rule is color, the correct choice would be the fourth card (4 blue circles) because it has blue shapes.
**Shape:** If the rule is shape, the correct choice would be the fourth card (4 blue circles) because it has circles.
Since I have no way of knowing the rule yet, I'm going to guess that the rule is based on color.
**Selection: 4**

⑨ **GPT-4o (CoT-TI) :**

To determine which of the four cards at the top the bottom left card belongs to, I need to consider the number of shapes, color of the shapes, and type of shapes individually, as these are the only possible criteria for the rule. The bottom left card has:- 1 blue circle.Now, let's analyze each card at the top to find the correct match based on each possible rule.
1. **Number of shapes**: - The bottom left card has 1 shape. - The first card at the top also has 1 shape (red triangle).
2. **Color of shapes**: - The bottom left card has blue shapes. - The fourth card at the top has blue shapes (4 blue circles).
3. **Shape type**: - The bottom left card has a circle. - The fourth card at the top has circles as well.
To determine which rule applies, I need to make an initial guess. Let's use the rule that matches my first identified characteristic: Since the bottom left card has 1 shape, I will use this as my first guess-Final Response: Selection: 1

✳ **Claude-3.5 Sonnet (CoT-TI) :**

Thank you for providing the detailed description of the image. Let's think through this step-by-step:
We have four cards at the top and one card at the bottom left. The bottom left card has 1 blue circle.
We need to match this card to one of the top four cards based on either number, color, or shape.
Looking at the **number**: Card 1 matches with 1 shape. Cards 2, 3, and 4 don't match in number.
Looking at the **color**: Card 4 matches with blue. Cards 1, 2, and 3 don't match in color.
Looking at the **shape**: Card 4 matches with circles. Cards 1, 2, and 3 don't match in shape.
Based on this analysis, we have two possible matches: Card 1 if the rule is based on the number of shapes. Card 4 if the rule is based on either color or shape.
Since we don't know the rule yet, and we're told it's based on only one characteristic, I'll make an educated guess and choose the card that matches two characteristics rather than just one.
Final response: Selection: 4

Figure B.5: WCST Task Example

**VISION_RESPONSE_EXAMPLE**

**Image :**

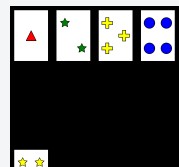

**Task Description :**

First, state how many cards you see in the image. Then, describe each card in detail. The cards contain shapes that vary in color, type, and number. For each card, provide the following information: number of shapes, color of shapes, and shape type. Format your response as: Number of cards: [number], Card Label: [number, color, shape], ...

✦ **Gemini-1.5 pro (CoT-VI) :**  ⊗

Number of cards: 6,
Card Label: 1 red triangle,
Card Label: 2 green stars,
Card Label: 3 yellow plus signs,
Card Label: 4 blue circles,
Card Label: 4 yellow stars,
Card Label: 4 yellow stars

⊛ **GPT-4o (CoT-VI) :**  ⊗

Number of cards: 6
Card 1: [1, red, triangle]
Card 2: [2, green, star]
Card 3: [3, yellow, plus sign]
Card 4: [4, blue, circle]
Card 5: [3, yellow, star]

✳ **Claude-3.5 Sonnet (CoT-VI) :**  ⊘

Number of cards: 5
Card 1: 1 red triangle
Card 2: 2 green stars
Card 3: 3 yellow plus signs
Card 4: 4 blue circles
Card 5: 4 yellow stars
Each card is depicted as a white rectangle on a black background, containing simple geometric shapes in various colors and quantities. The cards are arranged in a row at the top of the image, with the fifth card positioned below the others.

Figure B.6: WCST Vision Example

