# OpenReview forum: "Visual Large Language Models Exhibit Human-Level Cognitive Flexibility"
_ICLR.cc/2025/Conference — ICLR 2025 Conference Withdrawn Submission_

### Official Review · Reviewer_FXmB · 2024-10-20

**Soundness:** 2
**Presentation:** 3
**Contribution:** 2
**Rating:** 5
**Confidence:** 3

**Summary:**

The paper "Visual Large Language Models Exhibit Human-Level Cognitive Flexibility" evaluates the cognitive flexibility of state-of-the-art VLLMs (GPT-4o, Gemini-1.5 Pro, and Claude-3.5 Sonnet) using the Wisconsin Card Sorting Test (WCST). It finds that VLLMs can match/surpass human performance in adapting to changing rules, especially with chain-of-thought reasoning and text-based inputs.

Key contributions:
1. VLLMs demonstrate human-level cognitive flexibility, particularly with CoT prompting.
2. Performance significantly changes based on input  modality (text vs. visual) and prompting strategy.
4. VLLMs can simulate cognitive impairments, offering potential for modeling brain function.

The study suggests that VLLMs has some cognitive abilities and points to potential in advanced applications in AI and neuroscience.

**Strengths:**

Experiment methodology: The paper is methodically thorough, using a well-established cognitive flexibility test WCST and evaluating SOTA VLLMs.. The experimental design includes 4 different setups and 6 scoring functions. This enables a detailed comparison under varied conditions, providing some level of robustness to the findings. Including human participants as a comparative baseline grounds the findings in a relatable context.

Clarity: The paper is easy to follow and well-organized, with a clear explanation of the WCST, input modalities, and experimental conditions. The results are presented in detailed tables and figures, aiding in the understanding of model performance comparisons

Originality: The paper explores an area by applying well known congnitive test to assess performance  in VLLMs. It introduces a unique approach by examining how these models simulate cognitive impairments, adding some level of depth and innovation to the study.

**Weaknesses:**

Overstatement of Cognitive Flexibility Claims: Although the paper demonstrates that VLLMs can achieve human-level performance in the WCST under specific conditions, the claim that they exhibit human-like cognitive flexibility seems overstated. Cognitive flexibility in humans involves a broader spectrum of real-world applications and examined using multiple tests, and the findings are limited to a highly structured test. A more cautious interpretation of the results would strengthen the paper's scientific rigor.

Role-playing Cognitive Impairments Needs Validation: While simulating cognitive impairments through role-playing prompts is innovative, this method remains speculative without validation against clinical populations. The paper could improve by discussing potential methods for validating these simulated impairments against real-world data, making the findings more actionable and grounded in reality.

Insufficient Validation of Prompt Designs: While the paper employs CoT and STA prompting strategies, it does not fully explore the impact of different prompting setups or attempt to validate the prompts across varied task conditions and models. For example, the reliance on CoT prompting for achieving high performance raises questions about how much of the cognitive flexibility observed in VLLMs is genuinely attributable to their internal architecture versus the external aid provided by sophisticated prompts (which for some of the models are not advertised as the suggested approach).

**Questions:**

1. While the paper demonstrates that VLLMs can achieve human-level performance in the WCST, the broader claim of human-like cognitive flexibility seems to require more context. Could the authors clarify how they see the findings generalizing to real-world applications? Specifically, how do the authors view the limitations of the WCST in capturing the full spectrum of cognitive flexibility in humans, and do they plan to evaluate VLLMs using additional tests that capture a wider range of flexibility?

2. The paper heavily relies on a specific cot prompting approach to achieve high performance. Could the authors provide more details about how different prompting strategies and setups affect the models' performance across tasks? Specifically, does prompt wording changes the performance significantly?

3. The paper evaluates three SOTA models. How do the authors envision their findings generalizing to other models (includingn on VLLMs), especially those with different architectures or less advanced capabilities? Are there plans to extend the study to a broader range of models or to compare different architectural approaches to cognitive flexibility?

4. Cognitive flexibility in real-world settings often involves adapting to highly dynamic environments where rules are unclear and change rapidly. Do the authors have plans to test the models in more dynamic, less structured tasks where adaptability is required in real time?

---

> ### Author Response · Authors · 2024-11-19
> **To Reviewer FXmB**
>
> We sincerely thank you for taking the time to review this paper. We deeply appreciate positive comments on our methodological rigor, experimental design, and clarity of presentation. Below, we address your main concerns and questions.
>
> **1.Weaknesses 1, Question 1 and Question 4:**
>
> __Response:__
> Thank you for your thoughtful comment on the scope of our cognitive flexibility claims. We agree that cognitive flexibility encompasses a broader spectrum than what the WCST alone can measure. We will revise our manuscript to more carefully contextualize our findings, emphasizing that we have demonstrated human-level performance specifically in WCST-measured set-shifting, which represents one important component of cognitive flexibility. To better reflect this scope, we propose revising the title to "Visual Large Language Models Exhibit Human-Level Set-Shifting Abilities in the Wisconsin Card Sorting Test." Besides, we note that this concern aligns with Reviewer 9Pxs's Concern 2 and Reviewer PFJz's Concern 4. While there exist other cognitive flexibility measures like DCCS, IED, and cued switch tasks, we specifically chose WCST due to its unique advantages. Many alternative tests were designed for children or offer simplified paradigms compared to WCST, potentially limiting their discriminative power for evaluating VLLMs' capabilities. While real-world scenarios could provide ecological validity, they typically engage multiple cognitive processes simultaneously, making it challenging to isolate and measure cognitive flexibility specifically. WCST's well-established nature and extensive validation in cognitive neuroscience make it particularly suitable for our research aims.
>
> **2.Weaknesses 2**
>
> __Response:__
> We note that this concern aligns with Reviewer 9Pxs's Concern 1 and Reviewer PFJz's Concern 7. We would like to respectfully note that our study already establishes meaningful connections with clinical observations. As detailed in Section 4.5, our findings align with documented patterns in clinical literature: the models' simulated impairments correspond with established patient behaviors across different prefrontal dysfunctions, such as orbitofrontal damage (Stuss et al., 1983) and dorsolateral prefrontal cortex lesions (Stuss et al., 2000). While direct comparison with new clinical data would be valuable, collecting such data from patients with specific prefrontal impairments presents significant practical challenges. We believe our current approach of comparing simulated patterns with well-documented clinical findings provides meaningful insights while remaining feasible. We commit to expanding the analysis in Section 4.5 to provide more comprehensive connections with clinical literature.
>
> **3.Weaknesses 3 and Question 2**
>
> __Response:__
> Thank you for raising this important point about prompt design. We want to clarify that while VLLMs indeed show strong cognitive flexibility under CoT prompting but struggle with STA, the CoT prompt itself is remarkably simple - merely asking "Let's think step by step…" This minimal reconfiguration of the models' internal state enables successful task completion, similar to how children might initially struggle with WCST until they learn the appropriate thinking strategy. This suggests that the capability for cognitive flexibility exists within VLLMs' architecture and only needs appropriate elicitation to emerge. Regarding your specific question about prompt wording sensitivity, we have already investigated this in Section 4.4 (Impact of Explicit Rule Exclusivity), where we demonstrate that removing the explicit rule exclusivity constraint significantly impacts performance. This indicates that clear, complete, and exclusive task descriptions do influence performance. We would be happy to conduct additional experiments on prompt variations if you have specific suggestions in mind.
>
> **4.Question 3**
>
> __Response:__
> We appreciate this valuable point about model generalization. While we initially considered testing a broader range of models, including both open-source and proprietary ones with varying architectures and sizes, we encountered several technical constraints. The WCST task requires 64 continuous rounds of visual dialogue, which many models cannot support due to limitations in handling continuous visual conversations or restricted context windows. Some models are limited to single-image or few-shot visual interactions, while others may face context length constraints when dealing with multiple encoded images. These technical barriers prevented us from conducting comprehensive tests across a wider spectrum of models. However, as these limitations are gradually addressed through model improvements, we would be eager to extend our research to encompass a broader range of models and compare different architectural approaches to cognitive flexibility. We appreciate your suggestion and believe this expansion would provide valuable insights for future work.

---

### Official Review · Reviewer_PFJz · 2024-10-31

**Soundness:** 1
**Presentation:** 2
**Contribution:** 1
**Rating:** 3
**Confidence:** 5

**Summary:**

This work aims to evaluate the cognitive flexibility of vision language models (VLMs), using a classic task from the neuropsychological literature (the Wisconsin Card Sort Task). The authors conclude that, under certain conditions (depending on input modality and prompting technique), VLMs can display human-level flexibility. Experiments are also reported in which prompting is used to simulate neuropsychological impairment.

**Strengths:**

- This work employs a well validated task from the neuropsychological literature, potentially enabling a rich comparison with human cognition.
- The experiments investigate several state-of-the-art VLMs, increasing the robustness of the findings.

**Weaknesses:**

- Most importantly, the results are not diagnostic regarding the relative cognitive flexibility of VLMs/LLMs and humans. This is because the human participants are effectively at ceiling. In order to have a meaningful comparison, a version of the task (or a different task) would need to be identified where human performance was not at ceiling.
- No theoretical motivation is provided for investigating cognitive flexibility in LLMs / VLMs. It is noted that this is a well studied task in the neuropsychology literature, which is true, but this does not automatically yield theoretically important questions about LLMs / VLMs. It is also suggested that 'This investigation not only advances our understanding of VLLMs but also offers insights into the nature of cognitive flexibility itself,' but it is not clear what insights this work offers about cognitive flexibility.
- There is also no explicit motivation for studying VLMs in particular, as opposed to LLMs. Is there any particular reason why it is important to study these processes in the visual domain?
- The paper only includes experiments with a single task. Many more tasks and conditions would be needed to support the claims that are advanced in this paper.
- For tests of large-scale pretrained models such as LLMs and VLMs, it is also important to try and ensure that the tasks used for evaluation are not present in the model's training data. This is a concern here given the popularity of this task in the cognitive literature. One possible approach might be to also test an equivalent version of the task that uses different surface features, to ensure that performance does not depend on memorization (or pseudo-memorization).
- There are no statistical tests provided throughout the entire paper, although there are many statements about the differences between certain conditions. It is important to perform statistical tests to determine which of these differences are reliable.
- It is unclear what's learned from the experiments simulating neuropsychological impairment. There are some assertions about similarities to the pattern of behavior in certain patient populations, but very few references, and no direct comparison with human behavior. It would be ideal to have a direct comparison with behavior to support such claims.

**Questions:**

### Questions
- What is the theoretical motivation for studying cognitive flexibility in VLMs / LLMs?
- What is the theoretical motivation for studying cognitive flexibility in VLMs in particular? What does the visual domain add to such an evaluation?

### Suggestions
- The task should be modified so as to identify conditions where human performance is not at ceiling.
- More tasks should be investigated.
- Statistical tests should be included to support comparisons.
- A direct comparison with human behavior should be included for the experiments simulating neuropsychological impairment.

---

> ### Author Response · Authors · 2024-11-19
> **To Reviewer PFJz (Part-I)**
>
> We sincerely thank you for your thorough evaluation of our paper. Your comments have highlighted important areas for improvement. We address each of your main concerns below.
>
> **1.The results are not diagnostic regarding the relative cognitive flexibility of VLMs/LLMs and humans because human participants are at ceiling. To make a meaningful comparison, a task or variant where human performance is not at ceiling is needed.**
>
> __Response:__
> We appreciate the concern about ceiling effects. However, we would like to clarify that in our experiments, cognitively healthy humans achieved a Categories Completed (CC) score of 4.73 (0.45), while Claude-3.5 Sonnet under the CoT-TI condition reached 5.00 (0.00), actually surpassing human performance. As extensively documented in cognitive neuroscience literature over past decades, patients with cognitive impairments often show significantly degraded performance on WCST, sometimes being unable to complete the task at all. This suggests that while inability to complete WCST reliably indicates cognitive flexibility deficits, successful completion - as demonstrated by both humans and VLLMs in our study - indicates at minimum an absence of significant impairment. We acknowledge that identifying tasks where human performance is not at ceiling would provide additional insights into the upper bounds of cognitive flexibility. Given WCST's status as the gold standard for assessing cognitive flexibility impairments in humans, our results suggest VLLMs can achieve at least basic human-level cognitive flexibility without significant deficits. We welcome the your suggestions for additional tasks or variants that could better differentiate performance at higher levels of cognitive flexibility.
>
> **2.Theoretical Motivation for Investigating Cognitive Flexibility in LLMs/VLLMs**
>
> __Response:__
> We appreciate your raising this important point about theoretical motivation. Recent work has extensively probed various cognitive capabilities of large language models, with studies like Strachan et al. (2024) showing human-level performance in theory of mind tests, while Fatemi et al. (2024) revealed limitations in complex temporal reasoning. Given that cognitive flexibility is widely recognized as a crucial capability in cognitive science literature, with its impairment severely affecting daily functioning, we believe examining this capacity in VLLMs provides valuable insights into model capabilities and limitations. The WCST, as a well-validated measure of cognitive flexibility, offers a rigorous framework for this assessment. Prior to our work, it was unclear whether VLLMs could successfully complete the WCST or might exhibit patterns similar to humans with cognitive flexibility impairments. Moreover, while studying neural mechanisms of cognitive flexibility in human brains presents significant challenges, VLLMs offer unprecedented access to internal representations and activations, potentially providing new tools for understanding cognitive flexibility itself. We thank you for prompting us to clarify these important theoretical foundations.
>
> **3.Motivation for Studying VLMs vs. LLMs**
>
> __Response:__
> We appreciate your question about studying VLLMs specifically. Our motivation stems naturally from the fact that the traditional WCST for human subjects has always been administered using visual cards (varying in color, shape, and number) alongside verbal instructions. Thus, it is most natural to evaluate language models that can similarly process both visual information and verbal prompts using the same testing paradigm. While we acknowledge that the task can be transformed into pure text descriptions - which we indeed explored in our experiments - starting with visual inputs maintains consistency with the classical neuropsychological literature. This design choice enabled us to systematically compare both visual and textual modalities, providing insights into how these models handle cognitive flexibility tasks across different input formats.

---

> ### Author Response · Authors · 2024-11-19
> **To Reviewer PFJz (Part-II)**
>
> **4.Single Task Limitation**
>
> __Response:__
> We note that this concern aligns with Reviewer 9Pxs's Concern 2, and we appreciate your suggestion about incorporating additional tasks. While there exist other cognitive flexibility measures like DCCS, IED, and cued switch tasks, we specifically chose WCST due to its unique advantages. Many alternative tests were designed for children or offer simplified paradigms compared to WCST, potentially limiting their discriminative power for evaluating VLLMs' capabilities. While real-world scenarios could provide ecological validity, they typically engage multiple cognitive processes simultaneously, making it challenging to isolate and measure cognitive flexibility specifically. WCST's well-established nature and extensive validation in cognitive neuroscience make it particularly suitable for our research aims. Nevertheless, we would greatly welcome your suggestions for additional tasks that could assess cognitive flexibility with comparable precision to WCST, as this would certainly enrich our understanding of VLLMs' capabilities in this domain.
>
> **5.Training Data Contamination**
>
> __Response:__
> Thank you for the thoughtful concern about potential training data contamination. We would like to clarify that this potential issue was carefully considered during our experimental design phase - specifically, whether models might rely on memorization from similar tasks in their training data. However, we believe the nature of our task makes pure memorization highly unlikely, as each trial presents a unique challenge: the order of card presentation is entirely randomized, the sorting rules are randomly selected, and most importantly, the model must actively infer the current rule based on trial-by-trial feedback. Furthermore, rule switches are completely random, requiring real-time adjustment based on feedback after rule switches to ultimately discover the correct rule. Therefore, we believe this task does not rely on memorization. Of course, testing equivalent versions of the task using different surface features to ensure performance does not depend on memorization (or pseudo-memorization) could further demonstrate this point.
>
> **6.Lack of Statistical Testing**
>
> __Response:__
> We sincerely appreciate your rigorous suggestion regarding statistical testing. Your point about determining the reliability of differences between conditions is crucial. We will enhance our manuscript by incorporating comprehensive statistical analyses to validate these comparisons. Thank you for helping us strengthen the scientific rigor of our work.
>
> **7.Neuropsychological Impairment Simulations**
>
> __Response:__
> We note that this concern aligns with Reviewer 9Pxs's Concern 1, and we acknowledge that direct correlation with patient data would strengthen our findings. We would like to respectfully note that our study already establishes meaningful connections with clinical observations. As detailed in Section 4.5, our findings align with documented patterns in clinical literature: the models' simulated impairments correspond with established patient behaviors across different prefrontal dysfunctions, such as orbitofrontal damage (Stuss et al., 1983) and dorsolateral prefrontal cortex lesions (Stuss et al., 2000). While direct comparison with new clinical data would be valuable, collecting such data from patients with specific prefrontal impairments presents significant practical challenges. We believe our current approach of comparing simulated patterns with well-documented clinical findings provides meaningful insights while remaining feasible. We commit to expanding the analysis in Section 4.5 to provide more comprehensive connections with clinical literature.

---

> > ### Comment · Reviewer_PFJz · 2024-11-19
> >
> > I thank the authors for their replies. Unless the major issues with the current set of results and analyses can be addressed (e.g., ceiling effects, use of a single task, lack of statistical analyses, and lack of direct comparison to human data), I must keep my current score.

---

### Official Review · Reviewer_9Pxs · 2024-11-02

**Soundness:** 3
**Presentation:** 2
**Contribution:** 2
**Rating:** 3
**Confidence:** 3

**Summary:**

This article studies the cognitive flexibility of three multimodal large language models—Gemini, ChatGPT, and Claude—that support both text and image input using the WCST test. Cognitive flexibility here refers to the models' ability to adjust their understanding of task rules and complete tasks correctly based solely on feedback indicating correctness or incorrectness. The experiment includes SaT-VI, SaT-TI, CoT-VI, and CoT-TI conditions, where SaT means no chain-of-thought guidance and the model outputs answers directly, while CoT involves chain-of-thought guidance. The results show that CoT significantly outperforms SaT, achieving or surpassing human-level performance.

**Strengths:**

1.This study uses the WCST to examine the cognitive flexibility of VLLMs. The WCST is widely applied in cognitive science and is known for its strong reliability.2.The authors explored the potential of VLLMs to simulate specific patterns of cognitive impairment through role-playing.

**Weaknesses:**

1.Although the article mentions that the simulated patterns of the models align with real cases, the authors did not conduct cognitive experiments or correlate data with real subjects to demonstrate that VLLMs' simulation of cognitive impairment is reasonable.
2.The article only evaluates the models on a specific cognitive test (WCST). While the WCST is a classic test in cognitive science, it lacks real-world simulation, and performance on this test cannot fully represent performance in real-world scenarios.
3.The authors should consider incorporating more visualizations.

**Questions:**

Please see weaknesses.

---

> ### Author Response · Authors · 2024-11-19
> **To Reviewer 9Pxs**
>
> We sincerely appreciate the time and effort you dedicated to reviewing this paper and are grateful for your thoughtful and constructive feedback. Below, we address the main concerns you raised and look forward to receiving your further suggestions.
>
> **1. Concern: Lack of cognitive experiments and real subject data to demonstrate the reasonableness of VLLMs' simulation of cognitive impairment.**
>
> __Response:__
> We agree that correlation with real subject data would strengthen our findings and would like to respectfully note that our study does connect with clinical observations. As detailed in Section 4.5, our findings align with documented patterns in clinical literature: the models' simulated impairments correspond with established patient behaviors across different prefrontal dysfunctions, such as orbitofrontal damage (Stuss et al., 1983) and dorsolateral prefrontal cortex lesions (Stuss et al., 2000). While we agree that direct comparison with new clinical data would be valuable, collecting such data from patients with specific prefrontal impairments presents significant practical challenges. We believe our current approach of comparing simulated patterns with well-documented clinical findings provides meaningful insights while remaining feasible. We acknowledge that this comparison is currently limited to a brief discussion in the final paragraph of Section 4.5 and commit to expanding this analysis in the revised manuscript to provide more comprehensive connections with clinical literature.
>
> **2. Concern: Evaluation of models based on only one cognitive test (WCST), which may not fully represent performance in real-world scenarios.**
>
> __Response:__
> We appreciate your concern about the scope of cognitive testing. While there are indeed other tests for cognitive flexibility such as the Dimensional Change Card Sort (DCCS), Intra-Extra Dimensional Set Shift task (IED), and Cued switch tasks, some of these are either designed for children or are simpler than WCST. Real-world scenarios, while valuable, often involve multiple cognitive processes including working memory, inhibitory control, and reasoning, making it challenging to isolate and measure cognitive flexibility specifically. This is why we chose WCST, a well-established and extensively studied paradigm in cognitive neuroscience. However, we would greatly welcome your suggestions for any real-world tasks that could effectively assess cognitive flexibility with similar precision to WCST, as this would certainly enrich our understanding of VLLMs' capabilities in this domain.
>
> **3. Concern: Lack of visualizations in the paper.**
>
> __Response:__
> Thank you for the thoughtful suggestion regarding visualizations. We agree that additional visual representations could enhance intuitive understanding of our results. While we initially chose tabular presentations to ensure precise reporting of each metric, we appreciate your feedback and propose to convert Tables 2 and 3 into visual figures in the revised manuscript to improve readability while maintaining the same level of detail and precision. This modification would help readers better grasp the relationships and patterns in our findings.

---

> > ### Comment · Reviewer_9Pxs · 2024-11-20
> >
> > Thank the authors for their response. I did indeed notice the alignment analysis with human data in Section 4.5, but I believe a more detailed analysis is warranted here. Additionally, I agree with Reviewer PFJz's point regarding the motivation behind the analysis of VLLMs, which is not entirely convincing. Recent multimodal models still primarily use language models as their backbone, meaning that most of their reasoning and cognitive capabilities emerge from the language modality. As the authors also found, the model performed best under the CoT-TI condition. Introducing a new modality decoder may in fact reduce the model's cognitive and reasoning performance.
> >
> > Furthermore, I feel the authors have not provided sufficient evidence to demonstrate that the WCST test is appropriate for assessing the model's cognitive flexibility. As the authors correctly pointed out, "Real-world scenarios, while valuable, often involve multiple cognitive processes including working memory, inhibitory control, and reasoning, making it challenging to isolate and measure cognitive flexibility specifically." However, this very complexity makes it necessary to develop a reasonable approach to assess cognitive flexibility in real-world settings, and simply applying the WCST to models is not adequate. Therefore, I will not change my score.

---

### Note · Authors · 2025-01-28

I have read and agree with the venue's withdrawal policy on behalf of myself and my co-authors.